# Synergistic Effect of Dietary Supplementation with Sodium Butyrate, β-Glucan and Vitamins on Growth Performance, Cortisol Level, Intestinal Microbiome and Expression of Immune-Related Genes in Juvenile African Catfish (*Clarias gariepinus*)

**DOI:** 10.3390/ijms25094619

**Published:** 2024-04-24

**Authors:** Martyna Arciuch-Rutkowska, Joanna Nowosad, Łukasz Gil, Urszula Czarnik, Dariusz Kucharczyk

**Affiliations:** 1Department of Ichthyology and Aquaculture, Faculty of Animal Bioengineering, University of Warmia and Mazury in Olsztyn, Al. Warszawska 117A, 10-957 Olsztyn, Poland; martyna.arciuch@chemprof.pl (M.A.-R.); nowosad.joanna@gmail.com (J.N.); 2Department of Research and Development, Chemprof, Gutkowo 54B, 11-041 Olsztyn, Poland; lukasz.gil@chemprof.pl; 3Department of Ichthyology, Hydrobiology and Aquatic Ecology, National Inland Fisheries Research Institute, ul. M. Oczapowskiego 10, 10-719 Olsztyn, Poland; 4Department of Pig Breeding, Faculty of Animal Bioengineering, University of Warmia and Mazury in Olsztyn, ul. M. Oczapowskiego 5, 10-719 Olsztyn, Poland; czar@uwm.edu.pl

**Keywords:** breeding indicators, butyrate, immunity, microbiome community, prebiotics

## Abstract

The effect of dietary supplementation with sodium butyrate, β-glucan and vitamins (A, D3, E, K, C) on breeding indicators and immune parameters of juvenile African catfish was examined. The fish were fed with unenriched (group C) and enriched feed with a variable proportion of sodium butyrate/β-glucan, and constant content of vitamins (W1–W3). After the experiment, blood and the middle gut were collected. The microbiome of the gut was determined using Next Generation Sequencing (NGS). Liver tissue was collected for determination of expression of immune-related genes (HSP70, IL-1β, TNFα). W2 and W3 were characterized by the most favorable values of breeding indicators (*p* < 0.05). The highest blood cortisol concentration was in group C (71.25 ± 10.45 ng/mL), and significantly the lowest in W1 (46.03 ± 7.01 ng/ mL) (*p* < 0.05). The dominance of *Cetobacterium* was observed in all study groups, with the largest share in W3 (65.25%) and W1 (61.44%). Gene expression showed an increased number of HSP70 genes in W1. IL-1β and TNFα genes peaked at W3. The W3 variant turns out to be the most beneficial supplementation, due to the improvement of breeding and immunological parameters. The data obtained can be used to create a preparation for commercial use in the breeding of this species.

## 1. Introduction

In the last few decades, there has been a noticeable increase in the production of food from aquaculture. Aquaculture production in 1993 was approximately 41 million tonnes and in 2019—120 million tonnes [1]. Aquaculture has been identified by The Food and Agriculture Organization (FAO) as the fastest-growing food production sector [2]. Among the fish species with very high aquaculture potential is the African catfish (*Clarias gariepinus*) [3,4]. This species is very popular among consumers due to the taste and dietary properties of the meat. African catfish meat is characterized by a high content of digestible protein and omega-3 acids, with a low-fat content [5]. The increase in breeders’ interest in African catfish causes demand for specialized feeds, which the cost accounting for approximately 60% of all costs of commercial culture [6]. The composition of the feed, selected according to the needs of the fish species and developmental stage, contributes to a faster growth rate of the fish [7]. This shortens the approach time and reduces production costs. Therefore, many studies focus on developing improved fish feeding protocols to obtain the best breeding rates [8,9]. An important aspect of African catfish aquaculture is also improving immunity and reducing the occurrence of infectious diseases in breeding. This has a particular impact on the efficiency and profitability of breeding this species [10]. It is important to introduce non-antibiotic and non-hormonal growth promoters, immunostimulants and vitamins into the diet of African catfish, which will accelerate growth while improving the functioning of the immune system [11]. A review of the scientific literature shows that the above properties are characterized by, among others, organic acids in the form of salts, such as sodium butyrate and prebiotics, e.g., β-glucan, supporting the functioning of the intestinal microbiome [12,13].

Sodium butyrate is an organic compound belonging to the Short Chain Fatty Acid (SCFA) with a proven effect on promoting the growth rate of fish [14,15]. Numerous scientific studies have shown the positive effect of butyric acid and its sodium salt on the growth and development of fish such as European sea bass (*Dicentrarchus labrax*), sea bream (*Sparus aurata*), goldfish (*Carassius auratus*), carp (*Cyprinus carpio*) and Nile tilapia (*Oreochromis niloticus*) [16]. Sodium butyrate contained in the fish diet significantly improves feed intake while increasing the absorption and assimilation of nutrients, thereby increasing the final body weight and protein content in the carcass while reducing the lipid content [17,18,19]. This is because sodium butyrate increases the absorbent surface of fish intestines [20]. Sodium butyrate administered in the diet has a positive effect on the intestinal microbiome of fish, promoting probiotic bacteria *Lactobacillus* spp. while inhibiting the growth of pathogenic bacteria like *Aeromonas*, *Vibrio*, *Grammaproteobacteria* and coliforms [19,21]. Butyrate salts also participate in the regulation of proinflammatory cytokine gene expression. Supplementing the fish diet with sodium butyrate increases the expression of interleukin 1β (IL-1β) and tumor necrosis factor (TNFα) genes while inhibiting intestinal inflammation [22,23,24,25]. β-glucan is an organic compound belonging to polysaccharides, which is a component of dietary fiber. Due to its health-promoting properties, β-glucan has been classified as a prebiotic supporting the intestinal microflora of humans and animals [26]. There has also been interest in this nutrient in aquaculture. Due to its multiple health-promoting properties, such as antimicrobial, anti-inflammatory and antioxidant effects, β-glucan can be used in the diet as both a preventive and regenerative agent after infections [27]. Additionally, the effect of β-glucan on the innate immune response makes it an ideal compound to stimulate the immunity of fish [28]. β-glucan administered in the diet reduces the risk of *Aeromonas hydrophila* infection, affects the activation of antioxidant enzymes and reduces the risk of local intestinal inflammation [29]. β-glucan also has a proven effect on reducing the cortisol level of fish. As a result of long-term stress and, consequently, high levels of cortisol in the blood, immunosuppression may occur in the organism, which impairs the immune system. As a result of stress, fish are particularly vulnerable to infections [30]. Supplementing the fish diet with β-glucan can significantly inhibit and reduce the production of cortisol by adrenal cells [31].

It is also important to supplement the basic fish diet with vitamins [3]. This is justified by the fact that some vitamins are not synthesized in the body of fish or their synthesis is insufficient to meet the body’s needs [32]. Vitamins of particular importance for the proper development of the fish body are C (ascorbic acid), A (retinol), D3 (cholecalciferol), E (α-tocopherol) and K2 (menaquinone) [3]. Most commercial fish feeds contain inappropriate levels of vitamins, especially for species with very fast growth rates.

The combination of sodium butyrate with β-glucan in the nutrition of fish and other animals may enhance the health-promoting effects that they have as single ingredients by increasing the amount of SCFAs formed in the intestines. The fermentation of β-glucan conducted by the gut microbiota leads to the formation of mainly acetic, propionic and butyric acids. However, the dissociation of sodium butyrate in the gastrointestinal tract leads to the formation of butyric acid [33]. SCFAs produced because of metabolic and chemical changes become molecules mediating the interaction between intestinal microorganisms and the host, regulating many metabolic and immune processes [34]. The inclusion of all the above ingredients in the fish diet, i.e., sodium butyrate, β-glucan and vitamins, is an important issue because there are no studies that describe the synergistic effect of this composition on breeding indicators and immune parameters of fish. Preliminary research conducted on African catfish fry (0.48 ± 0.17 g; DAH = 26 days) showed a positive effect of feed supplementation with compositions containing the above ingredients on the growth performance, survival rate and intestinal microbiome profile of the tested fish [35].

The aim of this study was to investigate the effect of various compositions of sodium butyrate, β-glucan and vitamins (A, D3, E, K, C) provided in the diet on the breeding indicators and immune parameters of juvenile African catfish. For this purpose, growth indicators and survival rates were determined. Additionally, the profile of the intestinal microbiome, the level of cortisol in the blood and the expression of genes related to immunity (HSP70, IL-1β, TNFα) after a period of enriched feeding of juvenile African catfish were characterized.

## 2. Results

### 2.1. Growth Parameters

The highest values of growth parameters such as weight gain (WG), length gain (LG), specific growth rate (SGR) and Fulton’s condition factor (K) were determined in research groups W2 and W3 (*p* < 0.05). The control group (C) was characterized by the lowest values of growth parameters (*p* < 0.05). Statistical analysis showed no differences in SR between groups (*p* > 0.05) (Table 1). The change in average weight and length of the fish body during the feeding experiment is shown in Figure 1.

### 2.2. Cortisol Level in Blood Plasma

The control group was characterized by the highest concentration of cortisol in the blood of African catfish after the period of enriched feeding (71.25 ± 10.45 ng/mL) compared to the other groups (W1–W3). Statistically, the lowest cortisol level was determined in the W1 group—46.03 ± 7.01 ng/ mL (*p* < 0.05) (Figure 2).

### 2.3. Expression of Immune-Related Genes

After the enrichment time, increased expression of immune-related genes was observed (Figure 3). The highest level of HSP70 gene expression was determined in the W1 group compared to the other groups and the control group (*p* < 0.05). Statistically significant differences were also observed in the expression of IL-1β and TNFα genes, the highest levels were determined in the W3 group (*p* < 0.05).

### 2.4. Sequencing Data of Microbiome

Data from metagenomic sequencing of DNA from the intestinal contents of African catfish after a period of enriched feeding are summarized in Table 2. Group W1 and the control group (C) showed the highest number of observations, while groups W2 and W3 were characterized by the highest number of counts. The total number of amplicon sequence variants (ASVs) detected during the analysis was 514.

α-diversity indices such as the Simpson index, Shannon index, Evenness index and Chao–1 index were determined for all studied groups (Figure 4). The control group was characterized by the highest Simpson index value—0.65 ± 0.14, while its lowest value was determined in the W2 group—0.48 ± 0.17 (*p* < 0.05). Moreover, the Shannon and Evenness indexes also had the highest values in the control group: 1.78 ± 0.58 for the Shannon index and 0.09 ± 0.02 for the Evenness index, respectively. Group W2 was characterized by the lowest values of the above indices (*p* < 0.05). The control group (C) also showed the highest Chao-1 value, while the W2 group had the lowest (*p* > 0.05).

Principal coordinate analysis (PCoA) with the Bray–Curtis distance measure was used to determine the β-diversity of the intestinal microbiome of African catfish in all study groups after the period of enriched feeding (Figure 5).

The Venn diagram shows the number of common and unique ASVs for each group (Figure 6). A total of 112 common ASVs were determined, characteristic of all studied groups. The control group had the highest number of unique ASVs—111, while the W2 group had the lowest (29).

### 2.5. Profile of Gut Microbiome

Based on the obtained bioinformatics data, the intestinal microbiome profile of African catfish was determined after a period of enriched feeding. An analysis was carried out to determine the relative abundance of bacteria at the Phylum and Genus levels for individual groups. At the Phylum, *Fusobacteriota* dominated in all groups, from 61.44% in the W1 group to 51.26% in the control group (C). A high percentage of *Proteobacteria* was observed, the highest in group W2—31.76%, while the lowest in groups W1—14.22% and C—15.13%. The control group had the highest percentage of *Bacteroidota*—14.73%. The lowest percentage of this cluster was determined in the W2 group—5.74% (Figure 7, Appendix A). 

At the Genus level, the dominance of *Cetobacterium* was observed in all study groups, with the largest share in groups W3—65.25% and W1—61.44%. The control group had the lowest percentage of *Cetobacterium*. In group W2, *Rhodocyclaceae* of the C39 genus were determined in the amount of 29.04%, in the remaining groups their number was approximately 10%. In the intestinal microbiome community, the presence of *Barnesiellaceae* was found at the level of 7.78%, in the remaining groups W1–W3 their abundance ranged from 3.86% (W3) to 2.77% (W2) (Figure 8, Appendix A).

## 3. Discussion

This study examined the influence of three compositions of feed additives with different proportions of sodium butyrate: β-glucan and a constant amount of vitamins (A, D3, E, K, C) on growth performance, cortisol level in blood, and gene expression related to immunity (HSP70, IL-1β, TNFα), intestinal microbiome of juvenile African catfish. The obtained data show that supplementation of fish diet significantly improves the examined growth indicators compared to the control group. The highest values of WG, WL, SGR and K were observed in the W2 and W3 groups. Feed supplementation with sodium butyrate, β-glucan and vitamins in each of the research groups reduces the level of cortisol in the blood of African catfish. Gene expression analysis showed that feed supplementation increased the expression of IL-1β and TNFα genes in the W3 group and HSP70 genes in the W1 group.

One of the effects of using sodium butyrate is an increase in the absorbent surface of the intestine. These properties of sodium butyrate improve the absorption of active substances contained in food and their subsequent use by the organism [13,36]. This is justified by the results obtained in the presented paper. Group W2, which was fed with feed with the highest sodium butyrate content (150 mg/kg feed), showed the most favorable, statistically significant values of growth parameters such as WG, WL, SGR and K. Other authors, examining the influence of sodium butyrate contained in the diet on the growth rate of fish, obtained comparable results. Research by Zhao et al. [37] suggest a sodium butyrate dose of 500–1000 mg/kg of feed as the most beneficial in terms of stimulating the growth of yellow catfish (*Pelteobagrus fluvidraco*). However, Chen et al. [15] and Zhou et al. [38] in their research on the nutrition of fish such as largemouth bass (*Micropterus salmoides*) and juvenile golden pompano (*Trachinotus ovatus*) indicate 2000 mg/kg of feed as the optimal dose promoting growth. In the case of Pengze crucian carp (*Carassius auratus var pengze*), the dose of sodium butyrate causing the fastest weight gain of the tested fish is 2000–4000 mg/kg of feed [39]. Administration of sodium butyrate in the form of nanoparticles significantly reduces the most favorable dose-promoting growth rate. Studies conducted on Nile tilapia indicate that the content of 1.0–2.5 mg of sodium butyrate nanoparticles per 1 kg of feed affects the final weight and SGR of the fish [40,41]. Scientists also determine the upper limit of sodium butyrate content above which inhibits the growth rates of fish. El-Naby et al. [42] tested a commercial preparation (Gustor BP-70, Norel, Madrid, Spain) containing encapsulated sodium butyrate on Nile tilapia indicated 5000 mg/kg of feed as a favorable dose. However, they noticed a downward trend in weight gain and SGR above the mentioned dose. Scientists do not agree on the optimal dose of sodium butyrate in fish diets that positively affects breeding rates. This may indicate that the beneficial dose of this compound depends not only on the form of administration but also on the tested fish species and additional substances in dietary supplementation. This is confirmed by this study because the W3 group fed with feed containing sodium butyrate—50 mg/kg of feed and β-glucan—60 mg/kg of feed was also characterized by the statistically most favorable growth parameters (WG, WL, SGR and K) compared to the group W1 and the control group.

β-glucan used in aquaculture has properties that improve fish growth efficiency and strengthen the immune response to infections and inflammation. Aramli et al. [43], Do Huu et al. [44], and Khanjani et al. [45], examining the impact of β-glucan contained in fish diets on breeding indicators, noticed a similar relationship. According to the authors, a dose of β-glucan of 2000 mg/kg of feed causes an acceleration of the growth rate and SGR of fish such as juvenile Persian sturgeon (*Acipenser persicus*), pompano fish (*Trachinotus ovatus*) and rainbow trout (*Oncorhynchus mykiss*). Moreover, in the case of pompano fish and rainbow trout, a significant improvement in fish survival was found after a period of enriched feeding. Research conducted by Dou et al. [46] present no effect of dietary supplementation with β-glucan on the growth rate of Nile tilapia. However, it was found that β-glucan in the amount of 4000–5000 mg/kg of feed strengthens the immune response and resistance of the tested fish. Supplementation of β-glucan administered in the form of yeast cell walls to Pacific shrimp (*Litopenaeus vannamei*) also did not result in an acceleration of growth rate, but an improvement in immune parameters was demonstrated compared to the control group [47]. For this reason, β-glucan is perceived as an immunomodulatory substance, improving immunity and alleviating the symptoms of stress and inflammation [48]. Long-term high levels of cortisol in fish as a response to stressors, i.e., high salinity, low water oxygenation and noise, may lead to impaired immune response [30,49]. This condition, called immunosuppression, increases the susceptibility of the fish body to the development of bacterial and viral infections [50]. The scientific literature contains data confirming the thesis that β-glucan administered with the diet contributes to the reduction in cortisol secretion by intrarenal cells in fish [31,51]. The presented work also confirmed the above thesis. The control group (without supplementation) was characterized by the highest concentration of cortisol in the blood of the tested fish, while in group W1 the lowest level was observed, where β-glucan supplementation was at the level of 20 mg/kg of feed. However, the W3 group—fed with feed with the highest content of β-glucan, did not show any significant differences between the control group and the W2 group.

The tested preparation also included vitamins: C, A, D3, E and K, which are an important component of the fish diet due to the lack of or insufficient synthesis by the fish body [3,32]. One of the better-studied vitamins in the nutrition of African catfish is vitamin C. Its supplementation brings several benefits, such as faster weight gain and a healthy body experience [52]. Vitamin C deficiency in catfish causes skin darkening, reduced growth, flashing and erratic swimming [53]. Vitamin A at the appropriate level is necessary for the development of the skeletal system and proper vision in fish. Moreover, doses in the range of 800–1600 IU/kg of feed affect the proper growth and feed consumption of African catfish [54]. Vitamin D3 is especially important for fish due to its basic function of maintaining the proper development of the skeletal system. Fish, unlike humans, are unable to synthesize it in their bodies. The reason for this is probably the small amount of UV light that reaches the fish due to its shallow penetration in the water [55,56]. Vitamin E must also be supplied with the diet because fish do not have the ability to synthesize it. It is an antioxidant that protects macromolecules against oxidation. It has also been shown that its correct level in the diet has a positive effect on growth performance, survival and immune reactions [57]. However, the appropriate level of vitamin K contributes to the synthesis of blood coagulation factors and the regulation of calcium levels, thus influencing the proper development of the skeleton. The dose of required vitamin K in the diet depends on the species but has not been determined for African catfish [58].

The combination of sodium butyrate, β-glucan and vitamins in lower doses than recommended by other authors resulted in a significant improvement in the tested breeding indicators and immune parameters after nine weeks of enriched feeding of African catfish. The health-promoting effect of the above ingredients is probably caused by their synergistic effect. It is concluded that the SCFAs from β-glucan were supplemented with butyric acid derived from sodium butyrate. In this way, the pool of available SCFAs, especially butyric acid, was increased [33]. SCFAs play a key role in both nutrition and immunity as intermediary molecules between the gut microbiome and the host. SCFAs in the intestines primarily stabilize structures, ensure the integrity of epithelial cells of the intestinal mucosa and nourish colonocytes (mainly butyric acid). Moreover, SCFAs increase the secretion of antimicrobial proteins by the gastrointestinal epithelium, thus further strengthening the maintenance of intestinal balance [34]. The impact of SCFAs produced from β-glucan and sodium butyrate is not limited only to the digestive system. Through the circulatory system, SCFAs enter many organs and, by connecting with appropriate receptors, regulate their functioning, thereby influencing the immunity, inflammation and metabolism [59]. Although knowledge about the role of SCFAs in maintaining homeostasis in fish is less than in mammals, recent reports indicate the similarity of these mechanisms in both groups of animals [60]. It turns out that butyric acid derived from microbiota metabolism mediates increased resistance to infection in zebrafish through increased expression of IL-1β and an increase in the percentage of intestinal neutrophils [61]. Moreover, SCFAs (acetate, propionic and butyric acid) produced by fermentation of carbohydrates in the intestines can induce an increase in the antibacterial activity of head kidney macrophages isolated from turbot [62].

The literature describes the advantages of enriching fish diets directly with SCFAs in the form of improved growth performance and immunity [13]. However, it should be remembered that oligosaccharides including, among others, β-glucan are an important source of energy for the intestinal microbiome [63].

The composition of the intestinal microbiome of animals, including fish, plays a significant role in the functioning of the organism and building its immunity. Understanding the detailed relationships between the intestinal microbiome and the host may help solve the problems faced by the aquaculture industry. These include metabolic diseases, water quality, antibiotic and drug use [64]. Research leading to a detailed understanding of the fish intestinal microbiome and the relationships in the microbiome community will allow the creation of products that support the functioning of the fish [60]. In this study, the intestinal microbiome profile of African catfish was determined after a period of feeding with enriched food (W1–W3). Analysis of microbiome α-diversity showed the highest species diversity in the control and W3 groups and the lowest in the W2 group. This was also confirmed by the ASV analysis. The species diversity of the gut microbiome plays an important role in building immunity and fighting pathogens [65,66,67]. Research by Bozzi et al. [68] in Atlantic salmon (*Salmo salar* L.) suggests that the high biodiversity of the gut microbiome allows different microbial species to work together to counteract inflammation. The interaction of microorganisms influences the modulation of the immune response and increases the resistance to infection in the fish tested. Research on African catfish suggests that the introduction of a preparation combining sodium butyrate and β-glucan into the diet leads to an improvement in indicators such as the Shannon, Simpson and Chao-1 index compared to the control group [34]. This is an indication of an increase in the species richness of the microbiome under the influence of supplementation. It was also found that higher doses of β-glucan (60 mg/kg feed) increased the value of the α-diversity indices. However, higher doses of sodium butyrate (150 mg/kg diet) do not significantly improve the species richness of the microbiome examined, which was also found in this study [34]. The above statement is also confirmed by the study by Shon et al., 2024 where long-term administration of sodium butyrate to obese rats did not affect α-diversity indices. The difference in the effect of β-glucan and sodium butyrate on microbial diversity is due to the fact that sodium butyrate is not a source of energy for bacteria and is not metabolized by them. However, as a polysaccharide, β-glucan is fermented by microorganisms, thereby increasing the microbiome community [63].

However, the percentage of the main groups of microorganisms included in the microbiome community slightly differs between groups. According to Kim et al. [64] the main components of the microbiome of the fish are *Fusobacteriota* and *Proteobacteria*. *Fusobacteriota* is a particularly beneficial group of bacteria that produces butyric acid by fermenting carbohydrates [69]. Moreover, in both animal and human models, a positive role of *Fusobacteriota* during colorectal cancer is observed. *Fusobacteriota* activates inflammation to protect against pathogens that promote cancer development [70]. Group W1 had the highest percentage of this group—approximately 61%, while the control group and group W2 had the lowest—approximately 51%. At the Genus level, *Cetobacterium* dominated in all groups, the W3 group was characterized by the highest percentage of this genus—65%, while the control group had the lowest—51%. *Cetobacterium* deserves special attention due to its ability to produce vitamin B12 inside and outside the host [71]. This genus has also been observed in other freshwater fish species such as Nile tilapia [72] and *Cyprinus carpio* [73].

Pro-inflammatory cytokines, i.e., IL-1β and TNFα, play a key role in regulating the immune system of fish. Using cytokines as a specific marker, the level of the innate immune response under the influence of immunostimulators can be determined [74]. TNFα is one of the genes that is the first to be increased in expression in the early stage of fish infection, while IL-1β is responsible for the activation of lymphocytes and phagocytic cells [75,76]. In the presented paper, the highest level of IL-1β and TNFα gene expression was observed in fish from the W3 group. This group was fed with feed with the highest content of the immunostimulant—β-glucan. The immunomodulatory properties of β-glucan are caused by the interaction of the microorganisms nourished by it and intestinal epithelial cells. Improving host mucosal immunity results in the production of antimicrobial substances and mucosal immunoglobulins that modulate the inflammatory response [77]. Dawood et al. [12] and Rodriguez et al. [78] also reached similar conclusions about the effect of β-glucan on the expression of pro-inflammatory cytokines. Dietary administration of β-glucan to both Nile tilapia and zebrafish induced upregulation of IL-1β and TNFα. Heat Shock Proteins (HSP) are highly conserved chaperone proteins produced in large quantities by body cells exposed to stress factors, e.g., high/low temperature, toxins or infections. These proteins act as molecular chaperones and help protect cells against stress-induced damage [79]. HSP70 is the most frequently studied group of chaperones due to their involvement in many cellular processes, i.e., protein folding, transport across cell membranes and regulation of the response to heat shock [80]. They create a kind of buffer that allows cells to adapt to the state caused by external or internal factors [81]. Dawood et al. [12] studied the effect of β-glucan administered in the Nile tilapia diet (500 mg/kg feed) on HSP70 gene expression induced by different stocking densities. It turns out that β-glucan influences the modulation of HSP70 gene expression compared to unsupplemented groups with different densities. The presented work showed that the W1 group fed with feed with a β-glucan content of 20 mg/kg showed upregulation of HSP70 gene expression compared to the other groups. The W3 group with the highest β-glucan content (60 mg/kg feed) does not differ from the other groups in terms of HSP70 expression. A study conducted by Ran et al. [82] in Nile tilapia showed that dietary supplementation with baker’s yeast indirectly reduces HSP70 gene expression by reducing stress. This may confirm the lowest expression of HSP70 genes in the W3 group with the highest β-glucan content in the feed of the tested fish.

## 4. Materials and Methods

### 4.1. The Origin of the Experimental Fish

The fish used for the experiment were obtained from an African catfish breeding at the Department of Ichthyology and Aquaculture, University of Warmia and Mazury in Olsztyn. For this purpose, artificial reproduction was performed using Ovaprim (Syndel, Nanaimo, BC, Canada) at a dose of 0.5 mL/kg body weight. Reproduction, fertilization and incubation of eggs under controlled conditions were conducted in accordance with the methodology described by Kucharczyk et al. [83] and Abdel-Latif et al. [84].

### 4.2. Initial Rearing 

The larvae were raised to the juvenile stage in Recirculating Aquaculture Systems (RAS) at the Department of Ichthyology and Aquaculture, UWM in Olsztyn, according to the methodology described by Nowosad et al. [9]. During the first two weeks of rearing, the larvae were fed three times a day with live food in the form of freshly hatched brine shrimp (*Artemia* sp.) (Ocean Nutrition, Dartmouth, NS, Canada). After this time, complete feeds were introduced (Skretting, Nutreco, Stavanger, Norway), with granulation adapted to the fish’s body weight. During the entire initial rearing, the water temperature was maintained at 25 ± 0.1 °C (OxyGuard Pacific, Farum, Denmark). pH (Hanna HI 98128, Eden Way, Leighton Buzzard, UK), oxygenation, saturation (OxyGuard Pacific, Farum, Denmark) and the level of nitrates (Hach LCK339, Ames, IA, USA), nitrites (Hach LCK341, Ames, IA, USA) and ammonia (Hach LCK303, Ames, IA, USA) [85]. The tanks were cleaned of feces and uneaten feed every day. A total of 600 juveniles of African catfish with an average body weight of 258.4 ± 70.9 g and a length of 30.7 ± 2.6 cm were used for the present experiment.

### 4.3. Experimental Feed

To conduct the experiment, commercial complete feed Skretting (Nutreco, Optiline 3P, Stavanger, Norway) was used with the content of crude protein—41%, crude fat—22%, carbohydrates—19.1%, crude fiber—2.8%, raw ash—6.5% and total phosphorus—0.8%. This is the feed standardly used to feed African catfish in Polish commercial fish farms. The experimental feed was prepared in three variants (W1–W3) with different contents of active substances (sodium butyrate, β-glucan) and a constant content of vitamins (A, D3, E, K and C) per 1 kg of feed (Table 3).

The feed was enriched by adding an appropriate mixture (W1–W3) to the rapeseed oil: water mixture (1:1, *v*:*v*) in the proportion of 1:50 (*v*:*v*). The suspension prepared in this way was homogenized and injected into commercial feed (50 mL of suspension for each kilogram of feed). Feed was also prepared for the control group (C) enriched only with a mixture of oil and water (1:1, *v*:*v*) in the same proportion as in the research groups. The prepared feed was then dried at room temperature for 24 h [9,35].

### 4.4. Description of the Experiment

The experiment was conducted in Recirculating Aquaculture Systems (RAS) at the Department of Ichthyology and Aquaculture, University of Warmia and Mazury in Olsztyn. Fish were placed in 300-L breeding tanks 7 days in advance to acclimatize to experimental conditions. Each of the twelve tanks contained 50 fish. Each variant of the experiment was conducted in three repetitions (C, W1, W2 and W3). During the experiment, the water temperature was maintained at 25 ± 0.1 °C. Monitoring of water parameters was conducted in the same way as in the initial rearing. The fish were fed three times a day (8:00 a.m., 1:00 p.m., 5:00 p.m.) using automatic feeders. The daily dose was adjusted to the body weight and number of individuals [9,86]. The tanks were cleaned of feces and feed every day. The mortality in each tank of experience was also recorded. The experiment was conducted for nine weeks until the initial weight of the fish doubled. After the experiment was completed, the survival rate (SR; %) was calculated for each group with Equation (1) [87].
SR(%) = (Nf/Ni) × 100%(1)

SR—survival rate (%), Ni—number of fish at the beginning of the experiment, Nf—number of fish at the end of the experiment

### 4.5. Cyclic Measurement of Fish Body Weight and Length

Control measurements of the weight and length of the body of the fish were conducted at regular intervals (every three weeks) and at the end of the experiment. For this purpose, 30 fish from each variant were randomly caught and anesthetized using MS-222 solution with a concentration of 0.15 g/L (Sigma–Aldrich, Saint Louis, MO, USA). Body weight measurements were made on a scale with an accuracy of ±0.01 g (KERN ABJ; 440–49A, Balingen, Germany), and body length was measured using a caliper with an accuracy of ± 0.1 mm (Geko G01493, Radomsko, Poland). After the measurements, the fish were woken up in a tank with clean water and then transferred to the appropriate experimental tanks. The obtained measurement results were used to calculate growth indicators such as weight gain (WG; g), length gain (LG; cm), specific growth rate (SGR; %/d) and condition factor (K). The above parameters were calculated according to the Equations (2)–(5) [88].
WG (g) = FW − IW(2)

WG—weight gain (g), FW—final weight (g), IW—initial weight (g)
LG (cm) = FL − IL(3)

LG—length gain (cm), FL—final length (cm), IL—initial lenght (cm)
SGR (%/d) = (lnFW − lnIW)/T × 100%(4)

SGR—specific growth rate (%/d), FW—final weight (g), IW—initial weight (g), T—number of days of rearing.
K (g/cm^3^) = (BW × 100) × BL-3(5)

K—condition factor (g/cm^3^), BW—body weight (g), BL—body lenght (cm)

### 4.6. Sampling Procedure

At the end of the experiment, 10 fish from each group were randomly caught to collect samples for further analysis. The fish were placed in tanks with an anesthetic solution MS-222 at a concentration of 0.15 g/L (Sigma–Aldrich, Saint Louis, MO, USA). From the anesthetized fish, 2 mL of blood was collected from the tail vein into heparin tubes (FL Medical, Torreglia, Italy). The collected blood samples were centrifuged for 10 min at a speed of 10,000× *g* (IKA G-L S000, IKA, Staufen, Germany). The obtained plasma was stored at −80 °C (Labfreez LFZ-86L340, Beijing, China) until further analysis [89]. Then the fish from which blood was collected were sacrificed using an overdose of MS-222. The body surface of the fish was disinfected with a 70% isopropanol solution. The midgut was dissected under a laminar chamber (ESCO AC2-3E8-TU, Singapore) and its contents were collected into sterile Eppendorf tubes. Samples were protected and stored at −80 °C (Labfreez LFZ-86L340, Beijing, China) for further analysis [90]. Liver tissue was also collected and placed in RNAprotect Tissue Reagent (Qiagen, Hilden, Germany). The prepared samples were stored at −80 °C (Labfreez LFZ-86L340, Beijing, China) for further analysis.

### 4.7. Analysis of Cortisol Level in the Blood Plasma

The analysis of blood cortisol content was performed using enzyme-linked immunosorbent assay (ELISA) using the commercial Fish (Cortisol) ELISA Kit (SunRed Biological Technology, Shanghai, China). For this purpose, plasma samples were thawed and mixed using a vortex (V 3 S000, IKA, Staufen, Germany). The analysis was performed according to the manufacturer’s procedure using a microplate reader (BK-EL10C, BIOBASE, Jinan, China) at a wavelength of 450 nm.

### 4.8. RNA Extraction and Immune Related Genes Expression

RNA was isolated from liver tissue samples using the miRNeasy Tissue/Cells Advanced Mini Kit (Qiagen, Hilden, Germany) according to the manufacturer’s protocol. The integrity of RNA was verified by electrophoresis in a 1% agarose gel. The analysis of the quality and quantity of the isolated RNA was performed using a Nanodrop spectrophotometer (Thermo Scientific Nanodrop One, Waltham, MA, USA). The A260/280 ratio was in the acceptable range of 1.8–2.1. The isolated RNA was stored at −80 °C until gene expression analyzes began. Heat shock protein 70 (HSP70), interleukin 1β (IL-1β) and tumor necrosis factor alpha (TNFα) were selected to analyze the expression of genes related to immunity. The sequences of specific primers used for selected genes are presented in Table 4. RT-PCR (Reverse Transcription PCR) and real-time quantitative PCR (qRT-PCR) were performed in one step reaction using the QuantiNova SYBR Green RT-PCR Kit (Qiagen, Hilden, Germany) in a real-time PCR thermal cycler (AriaMx Real Time PCR, Agilent Technologies, Santa Clara, CA, USA). The RT-PCR and qRT-PCR were performed under the following conditions recommended by the manufacturer: reverse transcription—10 min/50 °C, PCR initial activation step—2 min/95 °C and 40 cycles of denaturation—5 s/95 °C and extension—10 s/60 °C. The composition of the reaction mixture was as follows: 10 μL RT PCR Master Mix, 0.2 μL RT-Mix, 1 uM forward and reverse primers, ≥200 ng of RNA temple and Rnase-free water to a total volume of 20 μL. Aria software (version number: 3.1.2306.0602) (Agilent Technologies, Santa Clara, CA, USA) was used to analyze the received data. The relative expression level of selected genes was determined using the ∆∆Ct method [91,92]. β-actin was used as a reference gene (internal standard) as suggested by Buwono et al. el. [93], Kari et al. [94], Nasrullah et al. [95] and Swaleh et al. [96]. β-actin gene was stable under various experimental conditions (*p* = 0.0655), which is also confirmed by Chaube et al. [97,98] in their research. The efficiency of primers was close to 100% (TNFα—92%, HSP70—96%, IL1β—112% and β-actin—102%). The linearity (r2) of the dilution series (6×) was not lower than 0.97 [99]. 

### 4.9. DNA Extraction, Metagenomic Sequencing and Bioinformatic Analysis of the Intestinal Microbiome

Isolation of DNA from the intestinal content of African catfish was performed using a commercial kit for DNA isolation from feces, QIAamp Fast DNA Stool Mini Kit (Qiagen, Hilden, Germany). DNA isolation was performed according to the manufacturer’s protocol. DNA quality control was performed using agarose electrophoresis on a 1% gel. The amount and purity of DNA were determined using a Nanodrop spectrophotometer (Thermo Scientific Nanodrop One, Waltham, MA, USA). After isolation, DNA samples were secured and stored at −80 °C until analysis (Labfreez LFZ-86L340, Changsha, China). NGS sequencing of the isolated DNA was performed by an external company (Genomed S.A, Warsaw, Poland) using the Illumina “16S Metagenomic Sequencing Library Preparation” protocol and the MiSeq Illumina system (Illumina, San Diego, CA, USA). The hypervariable V3–V4 region of the 16S rRNA gene was used to perform a metagenomic analysis of the microbial population. Primers 341F and 785R were used to amplify the selected region and prepare the library. PCR was performed using Q5 Hot Start High-Fidelity 2× Master Mix (New England Biolabs, Ipswich, MA, USA). PCR reaction conditions were used according to the manufacturer’s protocol. Sequencing was performed using paired-end (PE) technology, 2 × 300 nt, using the Illumina v3 kit (Illumina, San Diego, CA, USA) [35].

The QIIME 2 program was used to perform bioinformatics analysis classifying raw reads to the species level [101]. The DADA2 package was used to isolate sequences of biological origin and discard sequences generated during the sequencing process. Unique sequences of biological origin were also isolated (ASV—Amplicon Sequence Variant) [102]. The quality of the readings was checked using the FIGARO tool. Adapter sequences and too-short reads were removed. The obtained unique ASVs were assigned to appropriate taxa based on the Silva reference sequence database [101].

### 4.10. Statistical Analysis

Statistica 13.3 (TIBCO Software Inc.; Palo Alto, CA, USA) was used to perform statistical analyses. Normality and homogeneity of variances were analyzed using the Shapiro–Wilk and Leven tests. Breeding indicators (FW, FL, WG, LG, SGR and SR) and cortisol concentration were presented as mean with standard deviation (mean ± SD). For the statistical analysis of the above parameters, one-way ANOVA with Tuckey’s post-hoc test was used with a significance level of *p* < 0.05. The results from the bioinformatic analysis were present as the mean with standard error (mean ± SE). α-diversity indexes (Shannon index, Simpson index, and Eveness index) and Chao-1 index were calculated using the PAST 4.03 program (University of Oslo, Oslo, Norway). To conduct statistical analyzes of the expression of genes related to immunity (HSP70, IL-1β, TNFα), the non-parametric Kruscal–Wallis test was used with a significance level of *p* < 0.05. The results of the analyzes of common and unique ASVs were summarized in a Venn diagram (bioinformatics.psb.ugent.be, accessed on 15 September 2023). PCoA analysis (principal coordinate analysis) with the Bray–Curtis distance was used to present the β-diversity of the gut microbiome of all groups. In order to determine the profile of the intestinal microbiome of individual groups, the relative abundance (%) of bacteria at the Phylum and Genus levels was calculated and presented on bar charts.

## 5. Conclusions

The presented study shows that the simultaneous addition of sodium butyrate, β-glucan and vitamins (C, A, D3, E and K) to feed has a positive effect on both the breeding indicators and immune parameters of African catfish. This work proves that the use of active ingredients with a synergistic effect allows the use of lower doses of individual ingredients than recommended by other authors. SCFAs formed because of fermentation become mediators between the intestinal microbiome and the host, taking part in the regulation of many metabolic processes in the fish organism.

The ingredients used in the tested compositions can be successfully used in the production of complete feed or supplementary feed mixtures for juvenile African catfish. The impact of the above substances on spawners and the quality of the gametes they produce should also be examined. Moreover, to make the product more universal for use in aquaculture, the impact of the preparation on other species of farmed fish should be assessed.

## Figures and Tables

**Figure 1 ijms-25-04619-f001:**
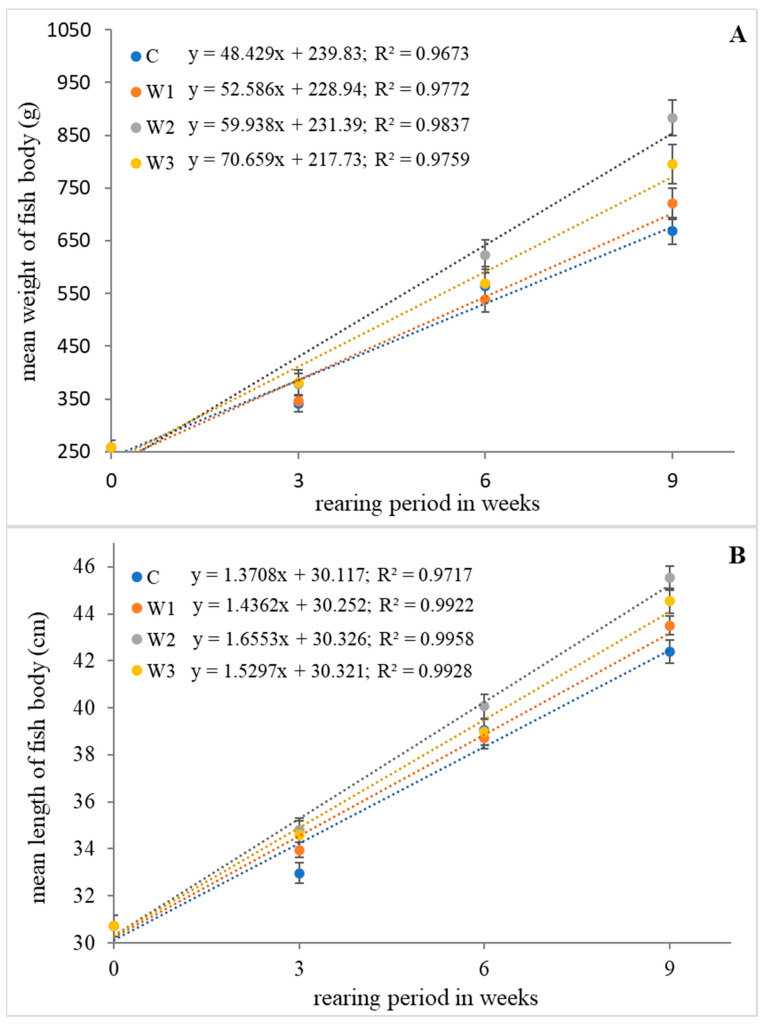
Relationship of weight (**A**) and length (**B**) of body of juvenile African catfish (*Clarias gariepinus*) fed commercial feed in the control group (C) and fed with enriched feed (W1–W3) in research groups during a feeding experiment. The error bars represent the standard error (*n* = 30) for each group.

**Figure 2 ijms-25-04619-f002:**
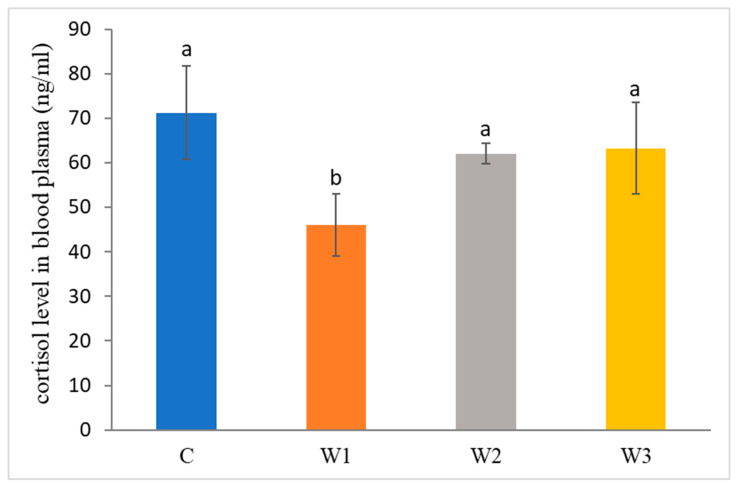
Cortisol level in blood plasma of juvenile African catfish (*Clarias gariepinus*) fed commercial feed in the control group (C) and fed with enriched feed (W1–W3) in research groups during a feeding experiment. Groups marked with different letters are statistically significant (*p* < 0.05). The error bars represent the standard deviations (*n* = 10) for each group.

**Figure 3 ijms-25-04619-f003:**
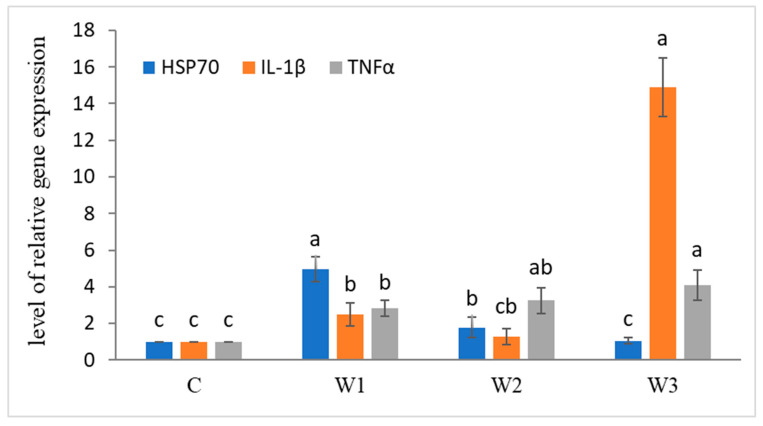
Expression of immune-related genes in liver tissue of juvenile African catfish (*Clarias gariepinus*) fed commercial feed in the control group (C) and fed with enriched feed (W1–W3) in research groups during a feeding experiment. Groups marked with different letters are statistically significant (*p* < 0.05). The error bars represent the standard deviations (*n* = 10) for each group.

**Figure 4 ijms-25-04619-f004:**
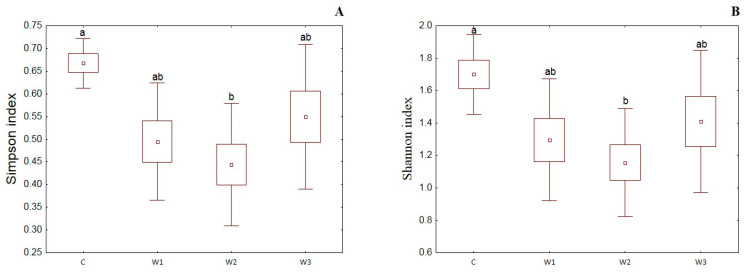
α-diversity metrics: Simpson index (**A**), Shannon index (**B**), Evenness index (**C**) and Chao–1 index (**D**) of intestinal microbiome of juvenile African catfish (*Clarias gariepinus*) fed commercial feed in the control group (C) and enriched feed (W1–W3) in research groups during a feeding experiment. Groups marked with different letters are statistically significant (*p* < 0.05). The error bars represent the standard deviations (*n* = 10) for each group.

**Figure 5 ijms-25-04619-f005:**
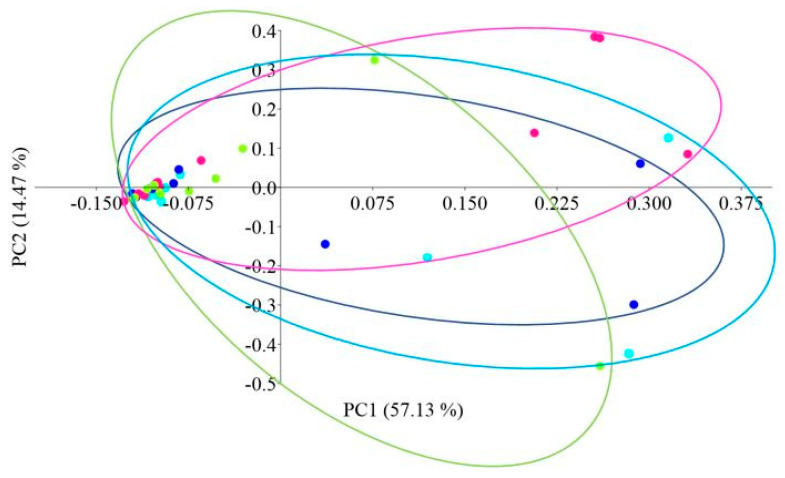
PCoA (Principal Coordinate Analysis) analysis using the Bray–Curtis distance of the intestinal microbiome of juvenile African catfish (*Clarias gariepinus*) fed commercial feed in the control group (C) and enriched feed (W1–W3) in research groups during a feeding experiment. Points of the same color mean individuals from the same group (C—light blue points, W1—dark blue points, W2—pink points and W3—green points).

**Figure 6 ijms-25-04619-f006:**
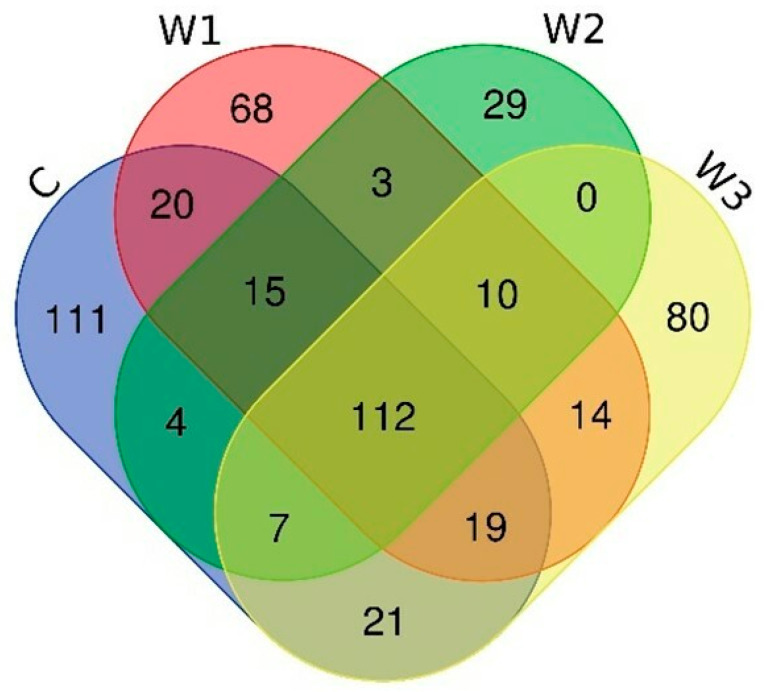
Venn diagram showing common and unique ASVs characterizing the intestinal microbiome of juvenile African catfish (*Clarias gariepinus*) fed commercial feed in the control group (C) and enriched feed (W1–W3) in research groups during a feeding experiment.

**Figure 7 ijms-25-04619-f007:**
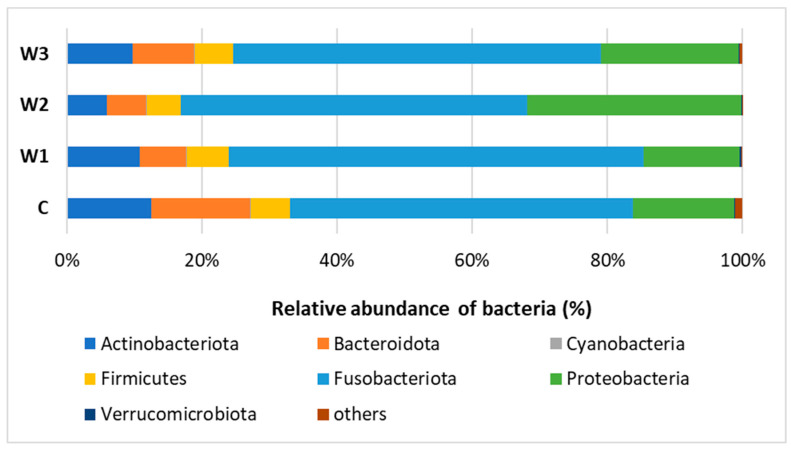
Relative abundance of bacteria at Phylum level in the gut microbiome of juvenile African catfish (*Clarias gariepinus*) fed commercial feed in the control group (C) and fed with enriched feed (W1–W3) in research groups during a feeding experiment.

**Figure 8 ijms-25-04619-f008:**
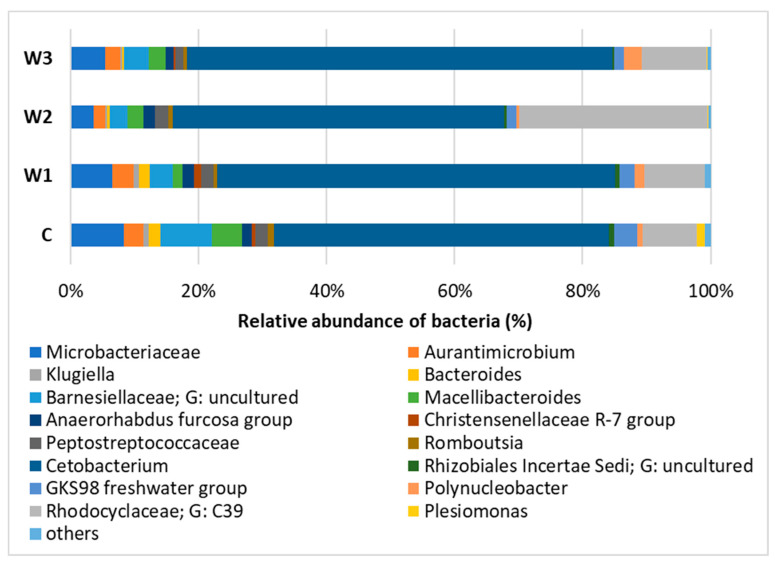
Relative abundance of bacteria at Genus level in the gut microbiome of juvenile African catfish (*Clarias gariepinus*) fed commercial feed in the control group (C) and fed with enriched feed (W1–W3) in research groups during a feeding experiment.

**Table 1 ijms-25-04619-t001:** Growth parameters (mean ± SD) of juvenile African catfish (*Clarias gariepinus*) fed commercial feed in the control group (C) and fed with enriched feed (W1–W3) in research groups during a feeding experiment. IW—initial weight, IL—initial length, FW—final weight, FL—final length, WG–weight gain, LG—length gain, SGR—specific growth rate, K—Fulton’s condition factor, SR—survival rate. Data in row marked with different letters are statistically significant (*p* < 0.05).

Growth Parameters	Groups			
	C	W1	W2	W3
IW (g)	258.37 ± 70.86			
IL (cm)	30.72 ± 2.60			
FW (g)	c 674.88 ± 147.44	b 751.21 ± 161.41	a 846.29 ± 182.25	a 840.24 ± 198.51
FL (cm)	c 42.62 ± 2.64	b 44.03 ± 2.49	a 45.36 ± 2.81	a 45.23 ± 2.88
WG (g)	c 416. 51 ± 147.44	b 492.84 ± 161.41	a 587.92 ± 182.25	a 581.87 ± 198.51
LG (cm)	c 11.90 ± 2.64	b 13.31 ± 2.49	a 14.64 ± 2.81	a 14.51 ± 2.88
SGR (%/d)	c 1.88 ± 0.24	b 1.99 ± 0.22	a 2.11 ± 0.21	a 2.10 ± 0.23
K (g/cm^3^)	b 0.86 ± 0.08	ab 0.87 ± 0.08	a 0.90 ± 0.13	a 0.89 ± 0.08
SR (%)	99.33 ± 1.15	99.33 ± 1.15	98.67 ± 1.15	99.33 ± 1.15

**Table 2 ijms-25-04619-t002:** Results (mean ± SE) of bioinformatic analyzes of intestinal microbiome of juvenile African catfish (*Clarias gariepinus*) fed commercial feed in the control group (C) and enriched feed (W1–W3) in research groups during a feeding experiment.

Groups	Observations	Counts
C	73.30 ± 13.76	71,891.20 ± 3860.08
W1	68.30 ± 13.35	80,012.90 ± 2251.85
W2	57.10 ± 6.63	85,198.20 ± 3295.04
W3	60.10 ± 15.34	85,028.30 ± 3073.70
Total number of counts taxonomically classified		3,221,306
Mean number of counts per sample		80,532.65 ± 1753.031
Total number of ASV		513

**Table 3 ijms-25-04619-t003:** Composition of the feed used during the feeding experiment on African catfish (*Clarias gariepinus*). C—control group, W1–W3—research groups.

Main Components	Groups			
	C	W1	W2	W3
crude protein (%)	39.05			
crude fat (%)	23.33			
carbohydrates (%)	18.19			
crude fiber (%)	2.70			
raw ash (%)	6.19			
phosphorus (%)	0.76			
Active ingredient				
sodium butyrate (mg/kg)	ND *	50	150	50
β- glucan (mg/kg)	ND *	20	20	60
vitamin C (mg/kg)	ND *	30	30	30
vitamin E (mg/kg)	ND *	10	10	10
vitamin K (mg/kg)	ND *	0.4	0.4	0.4
vitamin A (IU/kg)	ND *	1200	1200	1200
vitamin D3 (IU/kg)	ND *	800	800	800

* ND, no data.

**Table 4 ijms-25-04619-t004:** Primer sequence used for qRT-PCR.

Gene	Primer Sequences	Accession Number	References
β-actin	F: 5′ACCGGAGTCCATCACAATACCAGT 3′ R: 5′GAGCTGCGTGTTGCCCCTGAG 3′	KJ722167.1	[95]
HSP70	F: 5′CAAACGCAACACCACTATTCC 3′ R: 5′CATGGCTCTCTCACCTTCATAC 3′	MH341527.1	[95]
IL-1β	F: 5′TGCAGTGAATCCAAGAGCTACAGC 3′ R: 5′CCACCTTTCAGAGTGAATGCCAGC 3′	LC013677.1	[95]
TNFα	F: 5′TCTCAGGTCAATACAACCCGC 3′ R: 5′GAGGCCTTTGCGGAAAATCTTG 3′	KM593875	[100]

## Data Availability

The raw sequence from NGS microbiome used to support the findings of this study have been deposited in the RepOD repository (DOI: 10.18150/U6OXBT). Other data used to support the results of this study are available from the corresponding author upon request.

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
