# Peer review of "Synergistic Effect of Dietary Supplementation with Sodium Butyrate, β-Glucan and Vitamins on Growth Performance, Cortisol Level, Intestinal Microbiome and Expression of Immune-Related Genes in Juvenile African Catfish (Clarias gariepinus)"

_ijms, 2024, doi:10.3390/ijms25094619_

Round 1
Reviewer 1 Report
Comments and Suggestions for Authors
I read with pleasure and interest this article “Synergistic effect of dietary supplementation with sodium butyrate, β-glucan and vitamins on growth performance, cortisol level, intestinal microbiome and expression of immune-related genes in juvenile African catfish (Clarias gariepinus)”.
I must confess that a certain amount of information is lacking from the text. In item 44.8. RNA extraction and immune related genes expression, the ∆∆Ct method was employed. It is generally known that this method requires PCR reactions with near to 100% of efficiency, hence the efficiency of the reactions must be demonstrated. Furthermore, only b-actin was utilized as a reference gene; by the way, genes used for normalization should be referred to as reference genes, not housekeeping genes; it is not advised; instead, I suggest conducting a gene expression experiment in accordance with Bustin SA, Benes V, Garson JA, Hellemans J, Huggett J, Kubista M, Mueller R, Nolan T, Pfaffl MW, Shipley GL, Vandesompele J, Wittwer CT. The MIQE guidelines: minimum information for publication of quantitative real-time PCR experiments. Clin Chem. 2009 Apr;55(4):611-22. doi:10.1373/clinchem.2008.112797. So, at least the stability of b-actin expression across treatments must be demonstrated.
In terms of the figures, figures 1, 2, 3, 4, 5, 7, and 8 have poor definition; the numerals and letters are blurry, and the figures are visually polluted. It is possible to create good graphs using statistical software, thus I recommend creating new figures with a greater level of quality.
Author Response
Dear Reviewer
Once again, we would like to thank you for your valuable review, which will undoubtedly improve the final quality of our manuscript. All your comments have been taken into account by us. Detailed responses are included in the file attached to our response to your review.

Reviewer 2 Report
Comments and Suggestions for Authors
The manuscript entitled “Synergistic effect of dietary supplementation with sodium bu-tyrate, β-glucan and vitamins on growth performance, cortisol level, intestinal microbiome and expression of immune-related 4 genes in juvenile African catfish (Clarias gariepinus)” explore the potential effect of the simultaneous addition of sodium butyrate, β-glucan and vitamins (C, A, D3, E, K) on the breeding indicators and immune parameters of African catfish, which may serve as a new strategy for improving aquaculture performance in the future. However, there are still some problems should be revised. It is recommended that the authors refer to the following suggestions to further revise and improve the quality of the manuscript.
1.Abstract lacks a succinct summary of the research content.
2.I have some confusion about your experimental design. Is the study on the effects of vitamins on fish also part of this research? If so, why are there no ratio gradients set up in the experimental groups like sodium butyrate and β-glucan, and why is there no mention of the effects of vitamins on fish growth in the introduction section? If vitamins are not part of this study, why were vitamins not added to the control group and the same amount of vitamins added to all experimental groups?
3. It is recommended to adjust Figure 4 by removing the legend and marking the significant differences between groups.
4. Enlarge the size of the points in Figure 5 for better readability.
5. It is suggested to provide data tables corresponding to Figure 7 and Figure 8 in the supplementary materials, as the complex data cannot be effectively displayed in bar graphs to show all information.
6. Alpha diversity is a crucial parameter in the microbiome study. In the discussion section, Lines 329-333, you described the differences in alpha diversity between groups, but did not delve into a thorough discussion on these differences in conjunction with existing literature. Please supplement this section with relevant content, such as the relationship between additives and alpha diversity, as well as how additives may impact alpha diversity through various pathways.
Comments on the Quality of English LanguagePlease carefully check the spelling of the text and the format of the references in the manuscript.
Author Response

(The authors gave the same response as above.)

Reviewer 3 Report
Comments and Suggestions for Authors
Dear Authors,
I read the manuscript carefully. This MS contains very good results for fish farmers, especially catfish. There are few suggestions that can be applied in the text.
- Patterns should be used instead of colors for the columns of charts
- Statistically significant letters should be placed above the numbers in the tables.
Author Response

(The authors gave the same response as above.)

Round 2
Reviewer 1 Report
Comments and Suggestions for Authors
I am grateful to the authors for responding to the observations. The paper's authors included the efficiency data of the RT-qPCR reactions in the "4.8. RNA extraction and immune related gene expression" section, but they did not specify which pairs of primers had which efficiency. In any case, the efficiencies ranged from 92 to 112%, resulting in a 20% variation, which is considered quite significant and may influence the outcomes. Furthermore, it was neither justified nor demonstrated that b-actin expression did not vary among treatments.
Author Response
Dear Reviewer,
We would like to thank the Reviewer for the thorough analysis and proofreading of our manuscript. We tried to clarify all Reviewer's doubts. Our detailed response is presented in the attached file.
